# Association between Immunosuppressive Therapy Utilized in the Treatment of Autoimmune Disease or Transplant and Cancer Progression

**DOI:** 10.3390/biomedicines11010099

**Published:** 2022-12-30

**Authors:** Amanda Reyes, Atish Mohanty, Rebecca Pharaon, Erminia Massarelli

**Affiliations:** Department of Medical Oncology & Therapeutics Research, City of Hope National Medical Center, Duarte, CA 91010, USA

**Keywords:** autoimmunity, immunodeficiency, immunosurveillance, TNF-alpha inhibitors, calcineurin inhibitors, mTOR inhibitors, IMPDH inhibitors, purine antagonists, transplant, cancer progression, immune checkpoint inhibitors

## Abstract

Autoimmunity and cancer rates have both been on the rise in Western civilization prompting many to investigate the link between the two entities. This review will investigate the complex interactions between the activation and deactivation of the immune system and the development of malignancy. Additional focus will be placed on the main classes of immune inhibitor therapy utilized in transplant patients and in autoimmune disease including TNF-alpha, Calcineurin, mTOR, purine synthesis antagonists and IMPDH inhibitors.

## 1. Introduction

The concept of an immune system whereby innate cells identify and destroy foreign or malfunctioning cells has intrigued scientists for decades. As knowledge expanded so did the complexity of investigations. In the 1950s–1970s, researchers Burnet and Thomas formulated the concept of cancer immunosurveillance whereby a functioning immune system, at the time thought to be cells derived from the Thymus, can recognize ‘transformed’ cells or tumor cells [1,2]. However, there was limited proof of this concept in animal experimental models several decades after the initial theory was published [3,4,5]. More recently this concept of the protective immunosurveillance has been redefined into a broad topic of cancer immunoediting with 3 distinct phases termed elimination, equilibrium and escape [6] The phases covered variety of functions including both the innate and adaptive immunity as well as tumor recognition and tumor modification [6]. Additional support of this concept is demonstrated by the discovery of mice without properly functioning lymphocytes developed malignancy at a significantly higher rate compared to the wild type [7]. Thus, properly functioning immune systems are beneficial in the targeting and destruction of tumor cells.

Further investigation into the effects of upregulation of immune systems as seen in autoimmunity and downregulation seen in immunodeficiency/immunosuppression is needed. Autoimmunity and cancer rates have both been on the rise in Western civilization leading many to question if there is a link between the two entities. Likewise, the association between immunodeficiency or induced immunodeficiency and malignancy has been at the forefront of medical research with particular interest in the transplanted patients, HIV patients and patients with autoimmune disease requiring chronic immunosuppression. As the complexity and intricacy of immunosuppressive agents expands there is intensified interest in investigating down-stream effects of these agents

This paper aims to focus on the intersection of immune function and cancer with a review of the literature surrounding these two subjects. This is especially relevant in the age of exponentially increased use of immune checkpoint inhibitors in the treatment of cancers.

## 2. Autoimmunity and Cancer Correlation

Several studies have highlighted the connection between chronic inflammation and the development of malignancy [8]. While the exact mechanism of this relationship is not known, some speculate chronic inflammation leading to antigen specific cell damage, activation of Type 2 immune response mediated by IL-4 and IL-13, and relative CTLA-4 deficiency are potential triggers [8]. Other studies stipulate inflammatory cells themselves are a potential mechanism such as tumor-associated macrophages which can stimulate tumor growth and angiogenesis [9]. Observational representation of this concept can be seen in the increase in rates of autoimmune gastritis and its associated increase in rates of gastric cancer in younger women who have the highest incidence of autoimmune disease in comparison with a previously high prevalence of gastric cancer in men [10].

The association between autoimmunity induced inflammation and malignancy does not have a negative outcome always. A review by Zityogel et al. discussed the concept of ‘beneficial autoimmunity’ and identified examples of therapy-induced (such as immune checkpoint inhibitors) as well as idiopathic or spontaneous autoimmune disease conferring favorable outcomes against oncologic disease [11]. A higher level of T cells that recognize tumor-specific antigens available for anticancer immunosurveillance is thought to be another mechanism of beneficial autoimmunity [11]. In a large SEER [Surveillance, Epidemiology, and End Results program] database study, 13.5% of lung cancer patients were diagnosed with autoimmune disease during or after their cancer diagnosis [12]. However, a retrospective cohort study comparing lung cancer patients with and without autoimmune disease did not find any significant difference in progression-free survival per stage among the two groups, even though the autoimmune group was less likely to receive the standard of care inferring some protective benefit from the autoimmune disease [13]. In paraneoplastic encephalomyelitis, most commonly seen in patients with small cell lung cancer [SCLC], there was a favorable response to chemotherapy observed in patients who tested positive for anti-hu antibodies, autoantibodies against neuronal RNA binding proteins [14,15].

Given the evidence of beneficial autoimmunity, it would appear immunosuppression is a larger contributor to malignancy risk compared to autoimmune disease itself. Further evidence demonstrated in a 2020 review on celiac’s disease [treated with dietary modification] which found that overall malignancy risk was low and even negligible one year after diagnosis [16]. Likewise, a European study of autoimmune thyroiditis treated with hormone therapy did not find an elevated risk of thyroid cancer after a 10-year follow up period [17,18]. On the other hand, a Taiwanese study did find an increased risk of both thyroid cancer and colorectal cancer among Hashimoto’s thyroiditis patient population [19]. From this mixed data autoimmune disease must have a role in the development of malignancy.

Interestingly, malignancy has also been shown to stimulate autoimmunity, most notably through well-known paraneoplastic syndromes [20]. Paraneoplastic rheumatologic conditions such paraneoplastic polyarthritis and hypertrophic osteoarthropathy are a few notable examples and are thought be related to upregulation of VEGF and fibroblast growth factor 23 (FGF23) in tumors [20]. SCLC is another typical example of malignancy and associated with a high frequency of paraneoplastic syndromes that can proceed the diagnosis or be the first sign of relapse [21].

## 3. Immunodeficiency and Cancer Correlation

While overactive immune systems are a risk for malignancy, studies have also demonstrated that immunodeficiency may also be an independent risk factor [22]. This concept was first examined in mice models. In a research study, mice without an innate immune system were more vulnerable to both spontaneous and externally induced cancers [23]. A study involving the US Immune Deficiency Network Database examine primary immunodeficiency patients which included patients from 39 academic medical centers and >300 single gene mutations of the immune system in comparison to the age-adjusted SEER population. They identified a 1.42-fold increase in relative risk of malignancy in the primary immunodeficient population [22]. Interestingly, in subgroup analysis, men with primary immunodeficiency had the highest increase in relative risk, 1.91, compared to the age-adjusted population, while women were diagnosed with malignancy at similar rates [22].

Additional data provided by a large meta-analysis of HIV/AIDS patients [7 studies] and transplanted patients [5 studies] showed immunodeficiency and immunosuppression conferred a higher risk for malignancy in 20 out of the 28 types of cancer [24]. The authors proposed several mechanisms for the increased risk of malignancy including increased infections, immunodeficiency, increased cancer screening, and lifestyle factors [24]. However, the latter two theories were thought to be less likely as rates of screened cancers, breast and colon, were not elevated in both study populations and lifestyle factors were not standard across the study populations. Infection was thought to be an independent factor as many of the cancers in the analysis had an infectious cause [i.e., pylori, hepatitis, EBV] and several AIDS-defining malignancies were included [24]. However, notable exceptions were non-melanoma skin cancer and lip cancer that have no known infectious etiology and were found to have higher rates of malignancy in transplant recipients. Interestingly, the common epithelial-derived cancers, colon, breast, rectum, and ovarian, did not have higher rates among the immunosuppressed population [24].

## 4. Immunosuppression in Autoimmune Disease and Cancer Correlation

Autoimmune disease can be organ specific or systemic, but treatment and presentation vary greatly within each category. The mainstay of treatment in autoimmune disease can be separated into two categories, replacement therapy vs. immunosuppressive [25]. A NEJM review proposed that immunosuppressive treatment can be further characterized into 4 subtypes: alteration of thresholds of immune activation, modulation of antigen specific cells, reconstitution of the immune system and sparing of target organs [26]. Reconstitution of the immune system involves bone marrow ablation with or without the addition of stem cells, reserved for severe refractory disease as is an aggressive treatment [27]. More commonly used treatments target inflammatory pathways [26]. One such example are inhibitors of the TNF receptor as binding of its substrate triggers downstream signaling leading to upregulation of inflammation and apoptosis (Figure 1) [28].

###  Tumor Necrosis Factor-Alpha Inhibitors 

Tumor Necrosis Factor-Alpha [TNF-alpha] has been synonymous with pro-inflammation given its association with pro-inflammatory cytokines IL-1, IL-6, IL-8 and VEGF; therefore, its inhibitors are used for immunosuppression (Figure 2) [29]. However, there is some concern regarding the diverse functionality of TNF-alpha as low levels promoted angiogenesis while higher levels were anti-angiogenetic [30]. Many molecular pathways of TNF-alpha are associated with the upregulation of matrix metallopeptidase 9 [MMP9] which directly degrades the extracellular matrix allowing tumor migration and indirectly promotes cytokines that support cell tumor growth [31]. Despite this mixed data on TNF-alpha, researchers believed blocking its action could be a potential target for malignancy.

Thalidomide, an inhibitor of TNF-alpha protein synthesis (Figure 1), has been proven effective against certain cancer types including multiple myeloma, renal, breast, colon, and prostate among others, due to its inhibition of various growth factors, including VEGF, basic FGF, and Hepatocyte Growth Factor, as well as inhibiting tumor DNA synthesis [32,33]. In a clinical trial of 84 patients with refractory myeloma, 76 refractories to high-dose chemotherapy, were treated with escalating doses of Thalidomide to 800 mg over two months, the amount of myeloma protein determined response in serum or Bence Jones protein in urine [34]. Overall, there was a 32% response rate with greater than 90% reduction in 10 of the patients and the majority responded in two months. At 12 month follow up, overall survival was 58%, and event-free survival was 22% [34]. While the data on thalidomide is substantial, other medications in its class had significant adverse effects prompting further investigation. As a class, TNF-alpha inhibitors can promote lymphoproliferative disorders by an unclear mechanism [35]. One study by Askling et al. found an increased risk of leukemia and lymphoma in rheumatoid arthritis patients who were treated with TNF-alpha inhibitors [36]. Another found an increased risk of lymphoma in rheumatoid arthritis patients treated with etanercept or infliximab [37]. There is a dose-dependent relationship between TNF-alpha inhibitor use and the risk of malignancy [38]. Additionally, in a meta-analysis of 36 global clinical trials with 19,041 patients treated with adalimumab [Humira] for rheumatoid arthritis, ankylosing spondylitis, juvenile idiopathic arthritis, psoriatic arthritis, psoriasis, or Crohn’s disease had increased standardized incident ratio of non-melanoma skin cancer and lymphoma collectively [39]. With a few notable exceptions, caution should be used when treating patients with medications that modulate TNF-alpha.

Given this concern for malignancy risk, additional agents were developed with new targets of the inflammatory cascade; namely a monoclonal antibody targeting IL-12 and IL23 [40] and a4b7 integrin antagonist [41]. Contrastingly, both drugs [Ustekinumab and Vedolizumab] were not found to have an increased risk of new or recurrent malignancy in patients with IBD and history of previous malignancy [42,43].

## 5. Transplant Immunosuppression and Cancer Correlation

As mentioned above, immunodeficiency may be an independent risk factor for malignancy, affecting the transplanted population given the extensive immunosuppression to prevent rejection. In a cohort study using the US Scientific Registry of Transplant Recipients [SRTR], Non-Hodgkin’s lymphoma had the highest incidence apart from non-melanoma skin cancer among the most transplanted organs [kidney, liver, heart, lung] [Table 1] [44]. A study demonstrated that 1351 lung transplanted patients in Spain had higher rates of non-melanoma skin cancer [32%], lymphoproliferative diseases [18%] and lung cancer [16.5%] (Table 1) [45]. Further, another study of 2150 bone marrow transplant patients showed an elevated SIR of 11.6 of all cancer combined with the highest risk in b-cell lymphoproliferative disease [BLPD] [46]. In a more recent 2022 study on 2814 patients [only 23 met criteria for analysis] who received a liver transplant from 2008–2020 at a high-volume center found once again the highest risk of de novo development of non-melanoma skin cancer [21.7%] followed by gynecological cancers [17.3%] [47]. In addition, Non-Hodgkin’s lymphoma and Post-transplant lymphoproliferative disease were grouped separately unlike other studies in which they were grouped as the same entity [47].

Several studies have found that the higher rates of immunosuppression seen in heart transplanted patients conferred the highest rates of malignancy with skin cancer being the most common [48]. Additional data showed that post-kidney transplanted patients on immunosuppression had higher rates of lung cancer compared to rates prior to transplant while on dialysis [49]. Immunosuppression regimens can differ but commonly include calcineurin inhibitors, mTOR inhibitors, corticosteroids and anti-proliferative/antimetabolite [azathioprine]. According to the SRTR, the most common regimen used in 60–70% of kidney, heart, lung, liver, kidney-pancreas transplanted patients consist of tacrolimus, prednisone and mycophenolate mofetil [Table 1] [50]. While the risk of de novo malignancy after transplant has been noted in several studies, donor-derived malignancy is another entity.

In a highly publicized case from 2008, 4 patients who received organs from a 53-year-old donor with no known history of cancer developed molecularly similar breast cancer, confirmed by DNA microsatellites, within 16 months to 6 years of trans-plant [51]. Three of the four patients died from widely metastatic disease. Still the fourth patient, despite also having metastatic disease, was ultimately cured with removal of the transplanted organ and cessation of immunosuppression and chemotherapy [51]. While this was considered an infrequent event from estimates with a risk is 0.01–0.05%, it elicited further investigation into donor-derived malignancy [52]. A review from 2020 of 72 case reports, 50 case series, and six registries comprising of 234 patients found the most frequent donor-derived malignancy was lymphoma [20.5%], followed by renal cancer [17.9%], melanoma [17.1%], and non-small cell lung cancer [5.6%] [53]. Interestingly, in the analysis of the donor demographics, the donor had an ongoing history of malignancy in 17% of the cases while in 18.5% of the cases, the donors were not screened with imaging [53]. Whether the latent/dormant malignancies in donors evolved into aggressive disease as a consequence of immunosuppression alone or if there are other factors involved remains to be seen. Each Immunosuppressive agent requires further examination to evaluate malignancy risk.

### 5.1. Calcineurin Inhibitors

Calcineurin inhibitors, namely cyclosporine, have been employed in the transplant population for decades after the discovery of its selective inhibitory effect on T cells [54] (Figure 2). Later discovered to be the direct action of Cyclosporine A and Tacrolimus on calcineurin which uninhibited facilitates NF-AT translocation to the nucleus and activation of genes involved in the regulation of T cell proliferation (Figure 3) [55]. Popularity of this class of immunosuppressants increased after the observation of limited bone marrow suppression in animal models even at high doses [56].

Much research has been done to investigate the association between calcineurin inhibitors and their mechanism of action, prompting an investigation into potential malignancy concerns. A large meta-analysis on topical calcineurin inhibitors by Lam et al. did not find a significant increase in the risk of melanoma and keratinocyte carcinoma but identified a significant association with lymphoma. However, this was thought to be overall a minimal risk [57]. Systemic calcineurin inhibitors utilized in transplants, Cyclosporin and Tacrolimus, have been shown to promote tumorigenesis by increasing VEGF, TGF-B, and IL-6 in mice models [58]. In another large case–control study by Stewart et al., Cyclosporin A combined with other immunosuppressant medications, steroids and azathioprine, was associated with a decreased incidence of de novo breast cancer in female patients after kidney or heart transplants [59]. However, there was an increased incidence in all other types of malignancy reviewed compared to the ‘expected’ incidence obtained from an international data collection study in 1990s [59]. An additional investigation into a patient population of both sexes by the same team found a decreased incidence of rectal cancer among immunosuppressed patients but similar studies with prolonged follow-up intervals found an increased incidence [60]. From our current data, it remains difficult to form any unifying conclusions regarding calcineurin inhibitors and malignancy. Still, topical calcineurin inhibitors have a higher safety profile when compared to their systemic counterparts.

### 5.2. mTOR Inhibitors

MTOR inhibitors were first utilized in transplant in 1999 (Figure 2) [61] after the discovery of their activity blocking the proliferation of IL-2 stimulated T cells by halting the cell cycle progression from G1 to S [62]. Specifically, studies have demonstrated the binding and complex formation of Rapamycin with FKBP12 inhibits mTOR, most notably the mTORC1 portion (Figure 1) [63]. In addition, mTOR inhibitors have been shown to have a significant role in promoting T-cell anergy both in vitro [64] and in vivo studies [65]. Due to these functions, mTOR inhibitors remain an essential part of immunosuppression. Based on the literature, MTOR inhibitors play a more beneficial role regarding malignancy compared to other classes of immunosuppressants. Several in vitro studies have shown that rapamycin displays tumor targeting effects by triggering cell apoptosis (Figure 1) [66]. When rapamycin was used as a sole agent, it had the lowest incidence of skin cancers among the immunosuppressed population [67]. Additionally, rapamycin has been associated with decreased cancer risk when given without Cyclosporin A or when Cyclosporin A was withdrawn shortly after treatment initiation in renal transplant patients followed two years after the transplant [68]. However, the authors highlighted that short follow-up was a limitation of the study and additional studies with prolonged follow-up are needed to further quantify malignancy risk.

Chemical variations of the initial compounds were created to improve bioavailability and started being utilized to target and treat certain cancer types. The first, temsirolimus, an mTOR signaling inhibitor (Figure 2), was FDA approved in 2007 for advanced renal cell carcinoma after a phase III multicenter randomized open label trial of 626 previously untreated renal cell carcinoma patients with poor prognosis compared temsirolimus vs. temsirolimus plus IFN-alpha vs. IFN-alpha alone and found OS benefit of 10.9 months with temsirolimus monotherapy compared to 7.3 months with IFN-alpha alone [69]. Additional success was seen in mantle cell lymphoma but not in solid tumors [70]. One explanation is the cytostatic rather than cytotoxic effect of the drug which then induces cancer cell resistance rather than destruction [71]. To overcome this limitation, researchers combined the drugs with other proven effective cytotoxic agents, such as taxanes and carboplatin in ovarian cancer and metastatic melanoma [72,73]. A phase II of temsirolimus with carboplatin and paclitaxel followed by temsirolimus consolidation therapy in first-line stage III–IV clear cell carcinoma did not find statistically significantly improvement in progression-free survival. However, treatment was well tolerated and did find benefit in patients who were sufficiently debulked [72]. Less success was seen in the Alliance trial, a randomized phase 2 trial investigating carboplatin, paclitaxel, bevacizumab with and without everolimus in the treatment of stage IV metastatic melanoma, which found that the addition of everolimus was too toxic without additional survival benefit in comparison with the control arm, with progression free survival of 5 months in the 149 patients studied [73].

Everolimus was more successful when combined with endocrine therapy as it was shown to improve median progression-free survival with hormone therapy to 10.6 months compared to 4.1 months with hormone therapy alone in HR-positive breast cancer as second line in patients previously treated with aromatase inhibitors, according to the BOLERO-2, a phase 3 randomized trial of 724 patients [74]. Selective mTOR inhibitors were subsequentially developed to have a more targeted approach and in the hopes of achieving less toxicity and higher potency [75,76]. Several are currently in clinical trials. One such example is vistusertib (AZD2014), which is being evaluated in the treatment of recurrent grade II-III meningiomas (NCT03071874). Limitations to this drug class remain; a phase I clinical trial investigating BEZ235 in the treatment of advanced renal cell carcinoma (NCT00620594) was concluded early due to lack of efficacy and extensive toxicity associated with the drug with 50% of patients experiencing grade 3–4 toxicity [77]. Further, an additional randomized phase II trial of apitolisib, a combination mTOR/PI3K inhibitor in comparison with everolimus had shorter progression-free survival and a high rate of grade 3–4 toxicities [78]. Everolimus and temsirolimus have proven benefit in the treatment of select malignancies, combination mTOR and PI3K inhibitors should be monitored closely as toxicity is high.

## 6. Immunosuppression in Both Autoimmune Disease/Transplant and Cancer Correlation 

### Purine Synthesis Antagonists

IMPDH is an essential enzyme in the synthesis of purines, catalyzing the rate-limiting conversion of IMP to XMP, with the greatest impact in rapidly growing cells [79]. Mycophenolate mofetil (MMF), inhibitor of IMPDH, was first utilized as an immunosuppressant in rheumatologic disease (Figure 2) [80]. However, it is more commonly used in solid organ transplant in combination with calcineurin inhibitors [81]. The development of MMF stems from significant toxicity from Azathioprine and discovery of the selectivity of MMF for IMPDH in B and T cells [82,83]. Since this discovery, the utilization and interest in IMPDH inhibitors considerably increased.

In the late 1960s, researchers investigated the relationship of IMPDH inhibitors and anti-tumor properties; initial data was promising with anti-tumor activity against leukemia and sarcoma [84,85]. IMPDH inhibitors have been shown to decrease cell proliferation in several cancer types including leukemia, lymphoma, pancreatic, and non-small cell lung cancer [NSCLC] [86]. Unfortunately, this concept never made it past phase II clinical trials as effectiveness was limited at that time [87]. In a phase I dose escalation trial of mycophenolate mofetil in 11 relapsed and refractory multiple myeloma patients, disease progression was seen in 6/11 patients with stable disease in 4/11 and partial response in 1 patient [88]. In addition, severe GI side effects from direct enterocyte toxicity remained a limiting factor [89], as well as limited bioavailability due to degradation from glucuronidation [87].

As advancements in medical technology were made, methods were developed to overcome these shortcomings, such as the development of MPA nanofiber that could deliver a concentrated amount of the medication to a specific target for a sustained period in glioblastoma cells [90]. Additional investigation into the tumor targeting effects of IMPDH inhibition has identified benefits in tuberous sclerosis complex [91]. In addition to the benefits of IMPDH inhibitor’s tumor targeting effects, several studies have shown that transplant patients who received immunosuppression with IMPDH inhibitors, namely mycophenolate mofetil, had a decreased risk of malignancy including post-transplant lymphoproliferative disorders [92,93]. One proposed mechanism for the above finding is the IMPDH inhibitors’ antiviral properties exhibited against HIV and hepatitis, especially as an uncontrolled infection is associated with certain cancer tumorigenesis [94]. Additional research is currently underway to further explore molecular pathways of IMPDH and its downstream effects. Until such data exists to support both the efficacy and safety of IMPDH inhibitors in the treatment of malignancy, caution should be used prior to initiating treatment.

Antimetabolite Azathioprine, an additional antagonist of purine synthesis, has been utilized in the transplanted population for immunosuppression since 1960 (Figure 2) [95]. Its success as an immunosuppressant prompted further studies in the 1970s into its use in autoimmune disease, notably autoimmune hepatitis [96] where it has been proven beneficial in long-term remission after steroids have been withdrawn [97]. With its widespread and long-term use, questions of safety surfaced. A cohort study of patients with inflammatory bowel disease treated with azathioprine in Denmark from 1997–2008 found an increased overall cancer risk of 1.41 [95% CI 1.15–1.74] in particular lymphoid [2.42] and urinary tract cancer [2.84] [98]. Another investigation into the use of azathioprine and inflammatory bowel disease identified a higher risk of non-melanoma skin cancer with an odds ratio of 5.0 [95% CI of 1.1–22.8] [99]. Contrastingly, a meta-analysis of patients with myasthenia gravis on long-term azathioprine therapy did not find a significant increase in the risk of malignancy [100].

6-Mercaptopurine [6-MP], another antagonist of purine synthesis, was first used in the treatment of leukemias and related diseases in children in 1950s (Figure 2) [101]. More recently it was used as part of the consolidation protocol ALL2008 for pediatric acute lymphoblastic leukemia [NCT 00816049]. Like azathioprine, 6-MP was also utilized in the treatment of inflammatory bowel disease, but a review of 591 patients treated with 6-MP from 1969–1997 for an average of 5 years did not support higher malignancy risk [102]. A meta-analysis on both Azathioprine and 6-MP used in inflammatory bowel disease found an association with increased risk of lymphoma [non-Hodgkin’s], however, on further review, only one study of the six studies reviewed pertained to 6-MP, the others exclusively azathioprine [102,103]. Therefore, the data on 6-MP remains limited with the current studies and reviews.

## 7. Immune Checkpoint Inhibitors in Transplant and Autoimmune Disease

As the utility and wide distribution of immune checkpoint inhibitors [ICI] exponentially increased over the last decade, the question of their use in autoimmune disease and transplanted patients became more prominent. Given the concern for adverse effects, patients with autoimmune disease and transplanted patients have been excluded from the initial clinical trials limiting data [104]. This is further compounded by the fact there are no known methods or biomarkers to predict which patients will have severe immune related adverse events [105]. In an analysis of 49 publications and 129 patients with autoimmune disease, 75% had a flare of their pre-existing autoimmune disease while on ICI [106]. Interestingly, there was no difference in flares in active vs. inactive pre-existing disease [106]. A retrospective case series with 4438 patients, 283 who had autoimmune disease, found a significant increase in hospitalizations for immune related adverse events compared to patients without autoimmune disease [107]. However, a recent review found similar efficacy of ICI therapy in patients with pre-existing autoimmune disease compared to patients without [108]. There is some data that some patients with autoimmune disease benefit immensely from ICI treatment; in one case report, a patient with active Crohn’s disease requiring immunosuppression had a complete response of his metastatic melanoma with Pembrolizumab [109]. Less promising data was seen with ICI in solid organ transplants as the rate of allograft rejection was 41% in one review and 37% in another [110,111] but cause of death was more often from progression of cancer rather than transplanted organ failure [110]. Therefore, utilization of ICI in patients with autoimmune disease and solid organ transplants may be considered on a case-by-case basis with careful evaluation of risks and benefits.

## 8. Discussion

Based on the literature evidence, immunosuppressive treatment rather than autoimmune disease or transplant status confers a high risk for malignancy. Therefore, limiting the use may prove to be beneficial. Several of the investigations into the use of immunosuppressants in the transplanted population as well as the autoimmune group had long-term follow-up as many patients have been on immunosuppression for years. However, there was not distinction between continuous and intermittent use. If intermittent dosing is sufficient to prevent flairs and maintain disease stability, then patients may be spared at least a portion of malignancy risk. For patients on immunosuppression for decades, this potential modification may have exponential risk reduction. Additionally, for patients with both autoimmune disease and malignancy who might benefit from ICI therapy, alternative dosing of immunosuppressants may be optimal to facilitate this.

In compiling the many reviews and studies on transplanted patients, the highest risks overall were non-melanoma skin cancers [basal cell, squamous cell most common] and post-transplant lymphoproliferative disease [non-Hodgkin’s most common]. In addition, there is non-negligible risk of direct donor transmission of malignancy which may further complicate the matter and highlights the importance of strict donor screening protocols. Based on the US Scientific Registry of Transplants, tacrolimus with steroids and mycophenolate mofetil have been most commonly used in all transplanted solid organs by far for the last several years. This apparent standardization of immunosuppression may explain the similarity in cancer risk among the most transplanted organs. Not surprisingly, an elevated risk of cancer of the organ transplanted was noted except for heart transplants, likely related to the very low frequency of primary heart cancers. If another regimen of immunosuppression becomes utilized more frequently in the future, it would be interesting to evaluate the new regimen’s risk of malignancy and if any changes are seen. Current rates of graft rejection with ICI therapy in solid organ transplant are high but novel immunosuppressants may be able to mitigate this risk. Likewise, in the treatment of autoimmune disease, TNF-alpha inhibitors are at the center, but given the progression of targeted drug development these drugs may be replaced by alternative agents with their own set of risks.

Overall, various studies have demonstrated a direct association between malignancy and the immune system. It has also been determined that the relationship between malignancy and the immune system varies depending on whether the system is activated or deactivated. There are no absolutes regarding the downstream effects of overactivation or underactivation of the immune system as many factors are involved. The heterogeneity of malignancy itself complicates the matter further and makes drawing unifying conclusions from reviews and large studies difficult. Years of research have shown that certain immunosuppression agents carry a higher risk of malignancy while others have cancer targeting effects. This data may be helpful when selecting an immunosuppressive agent in transplant patients and patients with autoimmune disease. More research into the molecular intricacies of both malignancy and immunosuppression may help identify additional actionable connections and eventually potential therapeutic targets. Additionally, further investigation into this topic may reveal a potential biomarker or methodology to determine relative safety of ICI therapy in patients with autoimmune disease.

## Figures and Tables

**Figure 1 biomedicines-11-00099-f001:**
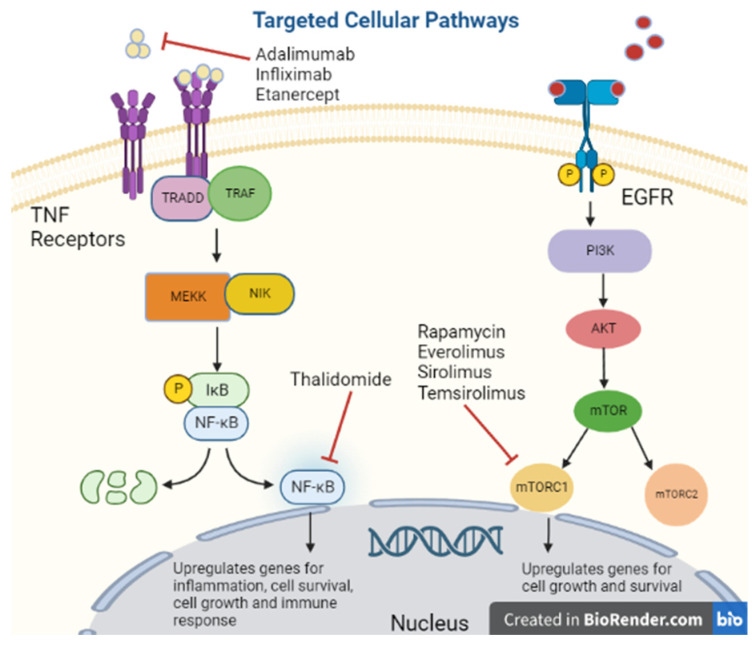
Targeted cellular pathways.

**Figure 2 biomedicines-11-00099-f002:**
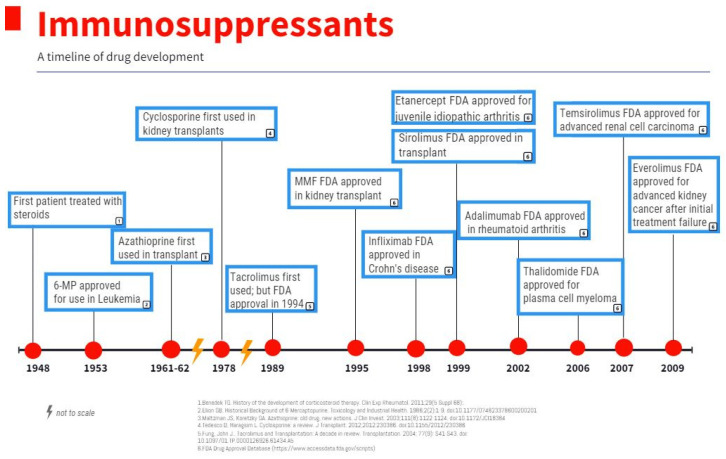
Immunosuppressants: a timeline of drug development.

**Figure 3 biomedicines-11-00099-f003:**
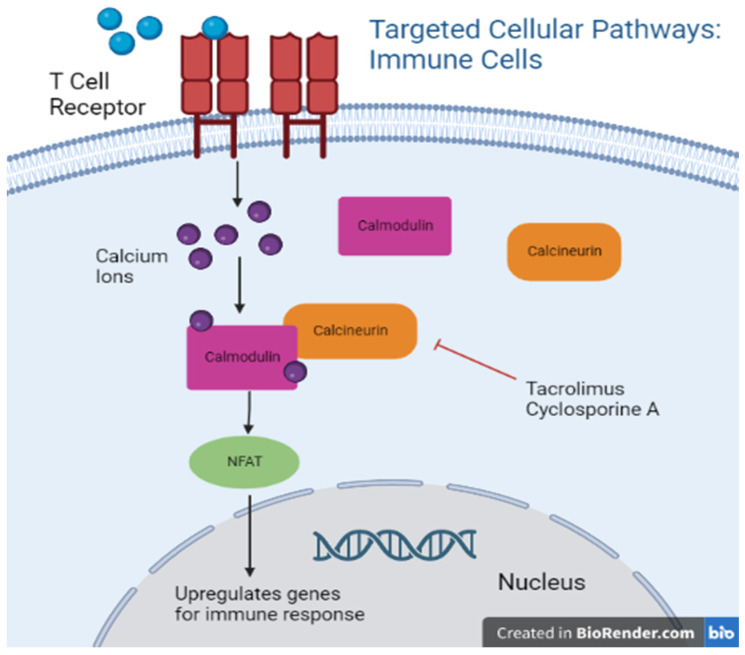
Targeted cellular pathways: immune cells.

**Table 1 biomedicines-11-00099-t001:** Organ transplants associated with highest cancer SIR.

Organ Transplanted	Highest Cancer SIRs ^1^	Most Common Suppression ^4^
Kidney	NMSC ^2^, PTLD ^3^, Kidney	Tacro + MMF + steroids
Liver	NMSC ^2^, PTLD ^3^, Liver	Tacro + MMF + steroids
Lung	NMSC ^2^, PTLD ^3^, Lung	Tacro + MMF + steroids
Heart	NMSC ^2^, PTLD ^3^, Lung	Tacro + MMF + steroids
Pancreas	NMSC ^2^, PTLD ^3^, Pancreas	Tacro + MMF + steroids

^1^ Standardized incident ratio ^2^ Non-melanoma Skin cancer ^3^ Post Transplant Lymphoproliferative disorder ^4^ Per the US Scientific Registry of Transplant Recipients 2020 data.

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
