# Peer review of "Association between Immunosuppressive Therapy Utilized in the Treatment of Autoimmune Disease or Transplant and Cancer Progression"

_biomedicines, 2022, doi:10.3390/biomedicines11010099_

Round 1

Reviewer 1 Report

This review explores the complex  interactions between the activation and deactivation of the immune system and the development of malignancy. It also highlights the possible risks of major immunosuppressants used in transplant patients and autoimmune diseases. This data may be helpful when selecting an immunosuppressive agent in transplant patients and patients with autoimmune disease.  Here is a small suggestion, it would be better if the author can summarize more relevant articles published in 2022 into this review.

Author Response

Given the historical nature of many of the drugs discussed, it was difficult to find articles from 2022 that were relevant to the topic of the review. 

Reviewer 2 Report

In this review, Reyes et al., have discussed the association between immunosuppressive therapy utilized in the treatment of autoimmune disease or organ transplantation and cancer risk. Although the content is descriptive but there are several points to be addressed to enrich it further. Overall, the article needs major revision before being considered for publication

Review paper Title: Association between immunosuppressive therapy utilized in 2 the
treatment of autoimmune disease or transplant and cancer 3 progression
Authors: Amanda Reyes, Atish Mohanty, Rebecca Pharaon, and Erminia Massarelli
Reviewer’s Comment
In this review, Reyes et al., have discussed the association between immunosuppressive therapy
utilized in the treatment of autoimmune disease or organ transplantation and cancer risk. Although
the content is descriptive but there are several points to be addressed to enrich it further. Overall,
the article needs major revision before being considered for publication.
1. The ‘Introduction’ section should be written more in detail.
2. Abbreviation of SEER?
3. In section 4, ‘Autoimmune Inhibitors’, need to describe briefly how they function or some
details.
4. In Figure 1, need to add reference for each drug developed mentioned in the timeline
5. In section 5, implication/importance of transplant immunosuppressants should be
described. What is the role of Calcineurin and it’s Inhibitors? Why it is used? Describe
clearly.
6. In section 5, although the authors described ‘mTOR inhibitors’ under ‘Transplant
Immunosuppressants’ but they discussed more here about it’s role regarding malignancy.
However, blockade of mTOR by rapamycin impairs dendritic cell (DC) maturation and
function, and inhibits T-cell proliferation, a mechanism that underpins its
immunosuppressive effect. There is strong evidence that mTOR is crucial for the regulation
of antigen responsiveness in CD4+ T cells. Here, the author could have put such important
information/studies about this aspect.
7. Section 6, what is IMPDH Inhibitors and it’s implication?
8. The author could have systemically classified some important autoimmune diseases like
rheumatoid arthritis (or classify important organ transplantation etc.) in different
section/subsection and describe the drugs used and the association with cancer.
9. The subtitles are not described in a well-organized manner.

Author Response

Dear Editors,

We thank you for the opportunity to resubmit a revised version of our manuscript titled “Association between immunosuppressive therapy utilized in the treatment of autoimmune disease or transplant and cancer progression” to Biomedicines.

We sincerely thank the reviewers for their thorough reading and constructive comments. We have carefully considered all their suggestions and have responded to the critiques, which have greatly helped strengthen our manuscript.

Reviewer #1: In this review, Reyes et al., have discussed the association between immunosuppressive therapy utilized in the treatment of autoimmune disease or organ transplantation and cancer risk. Although the content is descriptive but there are several points to be addressed to enrich it further. Overall, the article needs major revision before being considered for publication.

Comments:

  1. The ‘Introduction’ section should be written more in detail.

We have included more information on Immune system function particularly the importance of Immune surveillance as it pertains to the main topic of the review.

  1. Abbreviation of SEER?

We included the abbreviation of SEER (Surveillance, Epidemiology, and End Results program)

  1. In section 4, ‘Autoimmune Inhibitors’, need to describe briefly how they function or some details.

We added an entire paragraph briefly describing the generalized treatment approach immunosuppression versus replacement in autoimmune disease. We highlighted the categories of immunosuppression with emphasis on the mechanism of action of TNF-alpha inhibitors.

  1. In Figure 1, need to add reference for each drug developed mentioned in the timeline

We added references to the figure. The FDA approval data was obtained from the FDA database, the website was added to the reference data.

  1. In section 5, implication/importance of transplant immunosuppressants should be described. What is the role of Calcineurin and it’s Inhibitors? Why it is used? Describe

clearly.

We described both the role of calcineurin as well as the development of Calcineurin inhibitors. We also discuss the prominence of this drug class in transplants.

  1. In section 5, although the authors described ‘mTOR inhibitors’ under ‘Transplant Immunosuppressants’ but they discussed more here about it’s role regarding malignancy. However, blockade of mTOR by rapamycin impairs dendritic cell (DC) maturation and function, and inhibits T-cell proliferation, a mechanism that underpins its

immunosuppressive effect. There is strong evidence that mTOR is crucial for the regulation of antigen responsiveness in CD4+ T cells. Here, the author could have put such important information/studies about this aspect.

We added a paragraph describing the mechanism of action of mTOR inhibitors and the utilization in transplant. We included the role of mTOR in T cell anergy.

  1. Section 6, what is IMPDH Inhibitors and it’s implication?

We mentioned the function of IMPDH inhibitors and highlighted the most utilized compound MMF. As IMPDH inhibitors are antagonists of purine synthesis, was combined IMPDH inhibitors with Purine synthesis antagonists and entitled the subsection Purine Synthesis Antagonists instead.

  1. The author could have systemically classified some important autoimmune diseases like rheumatoid arthritis (or classify important organ transplantation etc.) in different section/subsection and describe the drugs used and the association with cancer.

We thank the reviewer for their comment. We have reformatted the sections for better readability.

  1. The subtitles are not described in a well-organized manner.

We reorganized the sections; Introduction, Autoimmunity and Cancer Correlation, Immunodeficiency and Cancer Correlation, Immunosuppression of Autoimmune disease and Cancer Correlation, Transplant Immunosuppression and Cancer Correlation, Immune Checkpoint Inhibitors in Transplant and Autoimmune disease and Discussion.

Thank you for considering this manuscript for publication. Your kind assistance in evaluating this paper is highly appreciated.

Sincerely,

Erminia Massarelli, M.D., Ph.D., M.S.

Associate Professor

Division Chief Thoracic Medical Oncology

Department of Medical Oncology & Therapeutics Research

City of Hope Cancer Center

Round 2

Reviewer 2 Report

All the question were answered, now it is qualified enough to publish this paper as it is. 

Author Response

Per reviewer, the comments have been addressed.